# A Review of Perovskite-Based Photodetectors and Their Applications

**DOI:** 10.3390/nano12244390

**Published:** 2022-12-09

**Authors:** Haiyan Wang, Yu Sun, Jin Chen, Fengchao Wang, Ruiyi Han, Canyun Zhang, Jinfang Kong, Lan Li, Jing Yang

**Affiliations:** College of Sciences, Shanghai Institute of Technology, 100 Haiquan Road, Shanghai 201418, China

**Keywords:** perovskite, photodetector, optical detection, lead-free

## Abstract

Perovskite photodetectors have attracted much research and attention because of their outstanding photoelectric characteristics, such as good light harvesting capability, excellent carrier migration behavior, tunable band gap, and so on. Recently, the reported studies mainly focus on materials synthesis, device structure design, interface engineering and physical mechanism analysis to improve the device characteristics, including stability, sensitivity, response speed, device noise, etc. This paper systematically summarizes the application fields and device structures of several perovskite photodetectors, including perovskite photoconductors, perovskite photodiodes, and perovskite phototransistors. Moreover, based on their molecular structure, 3D, 2D, 1D, and 0D perovskite photodetectors are introduced in detail. The research achievements and applications of perovskite photodetectors are summarized. Eventually, the future research directions and main challenges of perovskite photodetectors are prospected, and some possible solutions are proposed. The aim of the work is to provide a new thinking direction for further improving the performance of perovskite photodetectors.

## 1. Introduction

Photodetectors are effective devices that can convert optical signals into electrical signals. They are the eyes of the photoelectric system. Photodetectors have been widely applied to many fields. At present, most commercial photodetectors are made of silicon-based semiconductor materials and other inorganic semiconductor materials, such as gallium phosphide (molecular formula GaP), lead sulfide (molecular formula PbS), and so on. However, the complex and expensive manufacturing process under vacuum conditions is the tricky issue for the further development of the above commercial products. Polymer and quantum dot photodetectors are a new possibility in the field due to their low-cost solution fabrication process and broadly detected spectrum, covering from visible to near-infrared (NIR) range. Unfortunately, their low light absorption capacity and poor electric performance may be the fatal factors in their practical application.

In the last decade, due to their excellent electric characteristics (such as high carrier mobility, long carrier diffusion length), good optical characteristics (such as light absorption capacity, and tunable band gap) [1,2,3,4], and especially their simple and low-cost synthesis process under non-vacuum conditions, perovskite materials have become a promising candidate in photoelectric application. Until now, perovskites have been widely employed in optoelectronic devices such as solar cells [5,6], optical pump lasers [7,8], light-emitting diodes [9,10], and photoelectric detectors. Perovskites are ABX_3_ structure, where A is monovalent organic or inorganic cations (e.g., CH_3_NH_3_^+^, Cs^+^), B is inorganic cations (Pb^2+^ or Sn^2+^), C is halide anions (I^−^, Br^−^, Cl^−^). Geng et al. [11] integrated MAPbBr_3_ (MA = methylammonium) organic–inorganic perovskite single crystal (SC) material on a Si wafer to form a heterojunction photodetector with a detectable spectrum of 405–1064 nm. It had the advantages of fast response and high stability. In addition, it was combined with the imaging system to realize the typical gray-scale imaging of human faces, which is of great significance to face recognition in the future. In 2020, Huang’s group prepared a large area perovskite-filled membranes (PFMs) X-ray detector on a flexible substrate [12]. PFMs were formed by penetrating the porous polymer film with a saturated perovskite solution. Then, annealing and laminating were carried out. The sensitivity of the detector was 8696 ± 228 µC Gyair^−1^ cm^−2^ within the range of 0.05 V µm^−1^. Moreover, it did not degrade after being stored for more than 6 months under X-ray (376.8 Gyair), which prompted new ideas for the future research of X-ray photodetectors. 

In previous studies [13,14,15], the work focused on device structure design, interface engineering and fabrication approaches, and the physics mechanism of materials and devices has been reported. To enrich the understanding of perovskite photodetectors, we present a comprehensive overview of perovskite photodetectors here, which includes device types, device materials, practical applications (such as imaging, optical communication, biological detection fields) and effective improvement avenues for device performance (such as lead-free materials, stability). Firstly, three basic structures of perovskite photodetectors, as well as their working modes and basic principles, are discussed. In addition to the common three-dimension (3D) perovskite photodetectors, perovskite photodetectors with low-dimensional materials are summarized, including zero-dimensional (0D), one-dimensional (1D), and two-dimensional (2D) perovskite photodetectors. Thereafter, the application of perovskite photodetectors to self-powered photodetectors, imaging, biological detection, optical communication, up-conversion systems and polarized light are introduced. Then, for the lead-free and stability issues of perovskite, which have attracted much attention now, effective strategies for optimization and improvement are put forward. Finally, the outlook and future development directions of perovskite photodetectors are discussed. In addition, the roadmap of the current work is shown in Figure 1. 

## 2. Perovskite Photodetector Type

According to the device structure and working mechanism, perovskite photodetectors can be divided into three types: photoconductor, photodiode, and phototransistor. 

### 2.1. Perovskite Photoconductor

Figure 2 is the structure diagram of a perovskite photoconductive device. Metal electrodes were deposited on the perovskite surface to form a metal-semiconductor-metal (MSM) coplanar structure. The materials for metal electrodes include Au, Ag, Al, Cu and other metals. MSM structures absorb incident photons, and when the photon energy is greater than the band gap width (hv > Eg), the electrons of the MSM obtain enough energy to transit from the valence band to the conduction band, forming an electron–hole pair. When a bias voltage is applied at both ends of the device, the electron–hole pair is separated and collected by the metal electrodes through drift or diffusion, thus generating a photogenerated current. Dong et al. [16] deposited CsPbBr_3_ perovskite nanocrystals (CsPbBr_3_ NC_S_) on a Si substrate by centrifugal casting, and employed Au interdigital electrode pairs. The experiment found that the local surface plasmon resonance of all inorganic perovskite CsPbBr_3_ NC_S_ and noble metal nanocrystals overlapped well. This further improved the performance of the photodetector. Under the irradiation of a 532 nm laser with an illumination amplitude of 4.65 mW cm^−2^ and 2 V bias voltage, the switching ratio of the detector was greater than 10^6,^ and the dark current was 5 × 10^−4^ μA. The responsivity was 10.04 mA W^−1^ and the specific detectivity was 4.56 × 10^8^ Jones. 

Although perovskite photoconductors feature many appreciated advantages (such as simple structure, convenient preparation, the large photocurrent generated by the device and high responsivity), the distance between the two electrodes of a perovskite photoconductor is large, and part of the area of the semiconductor material is occupied by the metal electrode. Thus, the area of the active region is limited, which results in a relatively slow response speed, low sensitivity, and low specific detectivity. In addition, when the device works, a bias voltage is required to achieve stable photoelectric conversion. 

### 2.2. Perovskite Photodiode

Perovskite photodiodes are mainly built as p-i-n structure. In p-i-n structure, a wide intrinsic semiconductor (i layer) is employed to broaden the width of the barrier area of the p-n junction, extend the effective working area of the photoelectric conversion, form the p-i-n junction, reduce the p-n junction capacitance and improve the sensitivity of the diode. The schematic diagram of the p-i-n junction is shown in Figure 3. The principal structure diagram of the p-i-n type photodiode is shown in Figure 4a. Common p-i-n type photodiodes, due to their high cost, harsh preparation conditions and the limitations of their own characteristics, have nearly achieved the theoretical limit at their minimum size. Perovskite materials have good photoelectric and mechanical flexibility characteristics, which can be used to prepare flexible p-i-n type photodiodes. Figure 4b shows the structure diagram of the p-i-n type perovskite photodiode. Because p-i-n type perovskite photodiodes have electron transport layers (P region) and hole transport layers (N region), they can not only optimize the overall energy level matching relationship of the device, improve the carrier migration efficiency and reduce electron–hole recombination, but also inhibit the carrier injection from the electrode into the perovskite layer. Thus, it is favorable to reduce the generation of dark current. Furthermore, low-dimensional perovskite materials can also further reduce the device size and have excellent photoelectric properties. When the reverse bias voltage is applied to the perovskite photodiode, the voltage is mainly concentrated in the perovskite layer to drive the photogenerated carriers. Therefore, the carrier transit time decreases and the response speed of the photodiode becomes faster. 

Yang et al. [9] used organic–inorganic mixed CH_3_NH_3_Pb_3−x_Cl_x_ as a perovskite layer to prepare a p-i-n type perovskite photodiode. This device can detect weak light signals at room temperature, and featured a strong detection ability. The specific detectivity was close to 10^14^ Jones, the linear dynamic range (LDR) was more than 100 dB, and the 3 dB bandwidth was up to 3 MHz. On this basis, Lin et al. [10] prepared the perovskite photodiode structure of ITO/PEDOT:PSS/CH_3_NH_3_Pb_3−x_Cl_x_/PC60BM/C60/Ag/LiF. PC60BM and C60 could promote electron extraction and effectively prevent hole reverse injection. Thus, the dark current could be suppressed. Under the reverse voltage of −0.5 V, the minimum dark current of the device was 5 × 10^−10^ cm^−2^, LDR was 170 dB, the external quantum efficiency (EQE) was 50–70% in the wavelength range of 300~800 nm, and the 3 dB bandwidth was 500 kHz. Therefore, the above p-i-n structures were widely used for broadband optical detection in the visible region. A perovskite photodiode of ITO/ZnO Nanorods/CsPbBr_3_ quantum dots (CsPbBr_3_ QDs)/spiro-MeOTAD/Ag structure was prepared by Wang et al. [17]. The CsPbBr_3_ QDs completely penetrated into the interstices of ZnO nanorods to form a heterojunction structure. At 1.65 μW input laser power, with an external bias of 1 V, the device on-off ratio reached 3000, and the specific detectivity was 7.0 × 10^11^ Jones with the responsivity of 0.14 A W^−1^.

Perovskite photodiodes have excellent rectification characteristics. Compared with perovskite photoconductors, they have faster response speed and lower dark current, but lower responsivity and lower photocurrent.

### 2.3. Perovskite Phototransistors

Perovskite phototransistors are usually fabricated as three-end structures, such as a bipolar or field-effect transistor with three electrodes (source, drain, and gate electrodes). When the incident light illuminates onto the phototransistor, it is absorbed by the perovskite layer to generate photogenerated carriers. These carriers are transported to the individual electrodes by the action of the electric field between the source–drain electrodes. The three-end structure makes electrical controlling or electrical synchronization easier in perovskite phototransistors. The structure diagram of a perovskite phototransistor is shown in Figure 5.

Lee et al. [18] prepared CH_3_NH_3_PbI_3_ to integrate a perovskite/graphene heterostructure hybrid phototransistor into a silicon chip for the first time. Due to the high carrier mobility and low drive current of 2D graphene materials, the CH_3_NH_3_PbI_3_/graphene heterostructure could effectively reduce the recombination rate of photogenerated carriers and improved the electrical conductivity of CH_3_NH_3_PbI_3_ and the device responsivity. The responsivity of the device was 180 A W^−1^ and the EQE was about 5 × 10^4^%, while the detectivity was about 10^9^ Jones and the device had a wide spectral band in the UV-visible range. Chen et al. [19] prepared an ultrasensitive 2D-layered organic-inorganic hybrid perovskite (C_6_H_5_C_2_H_4_NH_3_)_2_SnI_4_ phototransistor which was applicable to visible application. The hole mobility of the channel layer reached 0.76~1.2 cm^2^ V^−1^s^−1^. When the incident light (447 nm, 5 μW cm^−2^) illuminated onto the phototransistor, the gate voltage was 16 V, the source–drain voltage was −16 V and the responsivity was 1.9 × 10^4^ A W^−1^. However, compared with perovskite photodiodes, perovskite phototransistors present a lower linear response to light intensity and time, higher dark current and slower light response speed. Table 1 illustrates the advantages and disadvantages of the parameters associated with perovskite photoconductors, photodiodes, and phototransistors.

## 3. Perovskite Photodetectors with Low-Dimensional Materials

On the molecular level, perovskite materials can be classified as 3D, 2D, 1D, and 0D materials. 3D perovskite has many advantages, such as longer carrier diffusion length, lower exciton binding energy and adjustable band gap; however, it is unstable in air and easily affected by temperature and humidity. Thus, an unexpected phase transition from a stable cubic phase to an orthogonal phase usually occurs under heating or in a moist environment [20,21]. 

Moreover, 3D perovskites are mostly polycrystalline films, which present a large number of grain boundaries and lattice defects, resulting in deep charge traps and surface traps. Moreover, the poor film is also attributed to the easy production of leakage current. Due to these limitations of 3D perovskites, attention is more often devoted to some low-dimensional perovskite materials, which can be formed by slicing 3D perovskite crystallographic planes [20]. 

### 3.1. 0D Perovskite Photodetector

0D perovskite is favored by researchers because of its strong exciton binding energy, increased radiative recombination in optoelectronic devices, rich structure, wide variety, and improved stability compared with 3D perovskite. 0D perovskites are composed of isolated metal halide octahedral anions or metal halide clusters, which are surrounded by organic or inorganic cations with excitons strongly confined to each octahedron [20]. 0D perovskite can be viewed as a core–shell structure, with negatively charged octahedra as the core, and organic and inorganic cations as the shell [20]. Since the octahedra of 0D perovskite are isolated, ion mobility can be effectively reduced, and the channels for ion mobility are disrupted. For example, while the light-absorbing layer (CsPbBr_3_ or MAPbBr_3_ perovskite) is coupled with the Ag electrode, the inhibition of ion migration can effectively avoid the combination of Br^−^ and Ag electrodes to generate AgBr precipitation, and destroy the optoelectronic performance of the device (as shown in Figure 6) [22]. Tang et al. prepared lead-free perovskite Cs_3_BiBr_6_ SC photodetectors on ITO conducting glass. The isolated BiBr_6_ polyhedra formed 0D Cs_3_BiBr_6_ perovskites [23]. The photodetector, illuminated at 400 nm, showed a good detectivity of 0.8 × 10^9^ Jones. The dark current was as low as 0.3 nA at a bias of 6 V. In addition, the Cs_3_BiBr_6_ crystals featured good stability under heating and a moist environment, as well the photodetector. In 2019, Zhang et al. [24] reported a deep UV light detector based on 0D leadless perovskite Cs_3_Cu_2_I_5_ crystal film. The results showed that the device was barely sensitive to visible light with a wavelength of 405 nm, but responded significantly to both deep UV 265 nm and UV 365 nm irradiation. The responsivity performance of the device decreased by only 7% when stored in an air environment for more than a month. The good stability of the device can be attributed to the unique structure of the 0D perovskite, in which the [Cu_2_I_5_]^3−^ octahedron in Cs_3_Cu_2_I_5_ was wrapped by the metal cation Cs^+^, which reduced the contact between water and oxygen in the air; thus, the optoelectronic characteristics of the device were more stable.

At present, the research on 0D perovskite photodetectors is still in its infancy. To improve the photoelectric performance of 0D perovskite photodetectors, it is necessary to investigate 0D perovskite itself. The dark current of the 0D perovskite photodetectors is significantly lower than that of the 3D perovskite photodetectors. Additionally, due to its special isolated structure, the temperature and moisture of the surrounding environment are isolated by the organic/inorganic cations wrapped in the outer layer. Thus, the stability is improved. Unfortunately, the characteristics of 0D perovskite, such as large band gap, high exciton binding energy, defects, and large Stokes shift [20] hinder its further development in optoelectronic devices. Therefore, some strategies need to be undertaken to adjust its band gap width and optical properties, so that 0D perovskite can be better used in optoelectronic detectors. Here are four potential research methods: (1) Substituting A-site/B-site cations to tune the band gap of 0D perovskite and optimize the photoluminescence (PL); (2) Enhancing the electronic coupling of inorganic inter clusters, designing 0D perovskites with a quantum well arrangement of inorganic clusters to form delocalized excitons, and then allowing the easy separation of excitons into free carriers [25,26]; (3) Replacing the electronic A-site organic cation with polar cations to decrease the dielectric mismatch between inorganic clusters and organic cations, reduce the dielectric limit, and obtain larger carrier mobility and lower exciton binding energy [20]; (4) Attempting to improve the optical properties of 0D perovskite, including ion doping [27,28,29], ligand engineering [30,31,32], the surface passivation package [33,34], and so on. 

### 3.2. 1D Perovskite Photodetector

Liu et al. [35] prepared a 1D perovskite (DME) PbBr_4_ SC phototransistor using antisolvent-assisted crystallization, in which DME was dimethyl ethylenediamine. The detector was exposed to ultraviolet light with a wavelength of 375 nm (1 μW cm^−2^) under the bias voltage of 20 V. The responsivity could reach 132.3 A W^−1^. Through the measurement of 1D (DME) PbBr_4_ SC by space-charge-limited current method, the minimum carrier mobility was 4.51 cm^2^/Vs, which was several times higher than the carrier mobility of 0.5 cm^2^/Vs of MAPbBr_3_ thin film field effect transistor [36]. However, it was much lower than the 164 ± 25 cm^2^/Vs mobility of millimeter-scale MAPbBr_3_ bulk crystals [37]. 

In 2020, Li et al. prepared 1D lead-free CsCu_2_I_3_ nanowire polarization-sensitive UV phototransistors by antisolvent-assisted crystallization [38]. CsCu_2_I_3_ nanowire had high crystallinity and strong emission anisotropy, and the photocurrent anisotropy ratio was about 3.16. The optical responsivity of the device was 32.3 A W^−1^ and the specific detectivity was 1.89 × 10^12^ Jones; light response time was 6.94/214 μs. In addition, the device was also prepared with flexible substrate polyethylene terephthalate (PET). It was found that after 1000 cycles of bending, the light response had almost no degradation. 

### 3.3. 2D Perovskite Photodetector

2D perovskite is a kind of 2D photoelectric material which is similar to graphene materials [18,19] and some new 2D materials, such as hexagonal boron nitride (h-BN) [39], transition-metal dichalcogenides (TMDs) [40,41,42], black phosphorus [43], and so on. They have the characteristics of excellent film forming, high carrier mobility, low exciton binding energy, strong light absorption capacity, adjustable band gap and high stability [44,45]. 

In 2D perovskite materials, due to the quantum confinement effect and the natural quantum well structure, electrons and holes are bound and can only move freely in two dimensions on the plane, perpendicular to the growth direction of the film. The preparation method of 2D perovskite involves introducing a large organic long-chain molecule into 3D perovskites, such as organic ammonium RNH_3_, to destroy the original cubic structure of the 3D material. The organic long-chain molecule separates the metal halide [BX_6_] octahedron of perovskite in a certain direction, such as along the [100], [110], or [111] crystal direction. The general chemical formula is (RNH_3_)_2_A_n−1_B_n_X_3n+1_. R is an organic functional group, A is an organic cation, B is a metal ion (generally Pb or Sn), X is a halogen anion and n is the number of octahedral layers stacked between two layers of macromolecules. When n = 1, it is 2D layered perovskite, when n = ∞, it is 3D perovskite, and when n is other values, it is quasi-2D layered perovskite. The [BX_6_] octahedron in 2D perovskite has three connection modes: common angle connection, common edge connection and coplanar connection [46]. Conventional 2D materials only show a quantum confinement effect when they are thinned to several atomic layers, meanwhile, 2D perovskites effectively prevent the entry of water and form a natural quantum well structure with quantum effect due to the isolation of [BX_6_] octahedron by long-chain molecular groups. Therefore, compared with 3D perovskites, 2D layered perovskites have better stability [47,48]. 

Zhang et al. [49] prepared a 2D perovskite Ca_2_Nb_3_O_10_ UV photodetector by calcination stripping method. The structure was quartz substrate/Ca_2_Nb_3_O_10_/Cr/Au, which was the perovskite photoconductive type. Under the detection wavelength of 270 nm and 1 V bias, the responsivity was up to 1156 A W^−1^ and EQE was 5.32 × 10^5^ %. The photodetector showed high performance at 3 V bias with 280 nm irradiation; the specific detectivity was 8.7 × 10^13^ Jones, the rising time of the device was 0.08 ms and the falling time was 5.6 ms. The UV/visible suppression ratio (R280/R400) of the UV photodetector was 8.84 × 10^3^, indicating that it had ultra-high spectral selectivity in the visible blind area. It was found that the performance of the device hardly changed when it was placed under the surrounding environment for more than 200 days, which indicated good stability. Combining Ca_2_Nb_3_O_10_ with PET, the detector showed excellent flexibility, and the LDR was 96 dB. Li et al. [50] synthesized centimeter-size, high-quality, pure-phase 2D perovskite heterostructures with tunable thickness and junction depth. They reported a heterojunction photodetector prepared by 2D layered perovskites. The structure of the photodetector was ITO/(C_4_H_9_NH_3_)_2_PbI_4_/(C_4_H_9_NH_3_)_2_(CH_3_NH_3_)Pb_2_I_7_/Cr/Au. At the wavelength of 540 nm with 45 mW cm^−2^ illumination, the responsivity of the detector was 0.69 A W^−1^, EQE was 158%, the rise and fall times of the device were 150 ms and 170 ms, separately, at a bias of −3 V.

Based on the self-trapping state in the 2D perovskite band gap and low out-of-plane conductivity, Li et al. [51] fabricated a perovskite photodetector with 2D (BA)_2_(MA)_n−1_Pb_n_Br_3n+1_ SC. The device structure was ITO/(BA)_2_(MA)_n−1_Pb_n_Br_3n+1_/Au, in which BA was C_4_H_9_NH_3_. Due to strong electron–phonon coupling, self-trapped states in the band gap of 2D layered perovskite were formed [52]. These self-trapping excitons had a long lifetime and weak optical transition intensity. The self-trapping phenomenon depended on the dimensions of the perovskites. Previous studies have shown that the self-trapping exciton of perovskite increases significantly from 3D to 2D [53]. Because of this characteristic, in 2D perovskite, the energy is lower than the band of free exciton absorption, and the self-trapped exciton will form an absorption starting point to assist absorption. The above realize the charge-narrowing collection effect of narrow-band optical detection through carrier recombination [53]. The working mechanism of narrow-band photodetectors fabricated by 2D perovskite was the coordination of charge-narrowing collection effect and self-trapping enhanced band gap absorption [51]. Illuminated under 620 nm irradiation (20 μW cm^−2^) with 5 V bias voltage, the rise and fall times of ITO/ (BA)_2_(MA) Pb_2_I_7_/Au detector were 125 ms and 74 ms, respectively, and the maximum specific detectivity was 1 × 10^11^ Jones, switching ratio up to 10^3^. In addition, the spectral response of the narrow-band photodetector could be continuously tuned from red light to blue light by changing the halide composition and the number of 2D perovskite layers n. Table 2 shows the comparison with the optical detection performance of low-dimensional perovskite photodetectors reported in the previous literature.

## 4. Application of Photodetectors

### 4.1. Introduction to Self-Driven Photodetector

Self-driven photodetectors bring hope for future independent, sustainable and wireless nanodevices and nano systems [54,55]. They can use solar cells to convert light energy into electric energy through the photovoltaic effect [5,56,57], acting as an external power source and driving perovskite photodetectors. They can also use the friction electrification effect [58,59] and piezoelectric effect to convert mechanical energy into electrical energy [60,61], convert bioenergy into electricity by using biofuels [62,63], and convert thermal energy into electrical energy by using the thermoelectric effect to realize the self-driven quality of the photodetector [64,65,66]. Lu et al. [33] integrated the solar cell and photodetector into a system. The device was only driven by its built-in electric field without external power supply. The work voltage was less than 1V and the sensitivity was high. Under periodic white light irradiation, the device still showed fast response speed and repeatability. In addition, the deposited ultra-thin alumina film on the perovskite layer of the photodetector, which played the role of protective layer, effectively improved the stability of the perovskite photodetector. This low-energy system provides a possible direction for saving energy, reducing the weight, and improving the portability of nano system equipment. Su et al. prepared a perovskite photodetector using the friction effect. When the external force allowed the upper copper electrode to make contact with the perovskite layer periodically and repeatedly, the friction charge, with opposite signs on the contact surface, caused the oscillating open circuit voltage (V_OC_) between the two electrodes [58]. When exposed to sunlight, MAPbI_3_ absorbs photons to produce electron–hole pairs, the mesoporous TiO_2_ layer captures electrons, and the holes neutralize the negative charges generated by friction on the contact surface. Therefore, when the photodetector receives light, the charge density of the contact surface decreases sharply. At the same time, the conductivity is improved, due to the generation of electron–hole pairs.

In addition to vertical self-powered photodetectors, Wang et al. [34] designed a self-driven transverse perovskite photodetector on rigid glass and flexible polyethylene phthalate substrate (PEN) based on CsPbI_3_-CsPbBr_3_ heterojunction nanowire array, via in-situ conversion and mask-assisted electrode manufacturing methods. It solved the problems of incident light loss in vertical devices and accumulation of a large number of defects on the interface. Its structure was a perovskite photoconductive detector. It provided a new concept for the preparation of heterojunction self-powered perovskite photodetector [34]. The perovskite layer adopted inorganic black phase CsPbI_3_ nanowire arrays and CsPbBr_3_ nanowire arrays. Inorganic black phase CsPbI_3_ was less affected by temperature, but it was easily affected by water and turned into a yellow phase (δ-Phase). Adding polyvinylpyrrolidone (PVP) molecule to CsPbI_3_ nanowires (NWs) could stabilize CsPbI_3_ crystal and change its surface energy state by interacting with C_S_^+^ [67]. At the same time, due to the 1D structure of the perovskite layer, the grain boundary could be significantly reduced and the detector was more stable. Under zero bias, the responsivity of the device was 125 mA W^−1^, and the rapid rise time and fall time was 0.7/0.8 ms. The optical response of the photodetector prepared on the PEN substrate was almost the same as that on glass. The device could maintain more than 90% of the initial performance under 500 bending cycles or a 180° bending angle.

In 2021, Liu et al. [68] prepared triple-cation mixed-halide perovskite FA_0.9_MA_0.05_Cs_0.05_PbI_2.7_Br_0.3_ single crystals (FAMACs SCs) (FA = formamidine). The use of the reducing agent formic acid (FAH) effectively reduced the phase separation resulting from iodide oxidation and cation deprotonation to I_2_ and I_3_^−^, avoided γ-CsPbI_3_ and formation of the γ-FAPbI_3_ hetero phase, and obtained the state-of-the-art perovskite SC with 8.74 ± 0.62 μs long carrier lifetime, 219 ± 18 cm^2^ V^−1^s^−1^ high carrier mobility, 71 ± 5 μm long carrier diffusion distance, superior homogeneity and long-term stability. A self-powered integrated circuits (SP-ICs) photodetector was fabricated by integrating SC photodetector arrays of FAMACs with a solar cell of Au/FAMACs SC/C60/2,9-dimethyl-4,7-diphenyl-1,10-phenanthroline (BCP)/Au structure using 12-bit interdigitated gold electrode integration between photodetectors. With different numbers of transversely structured perovskite SC solar cells in series, a larger bias could be obtained to drive the detector arrays. The SP-ICs photodetector exhibited good response, light conduction gain, specific detectivity and rapid photoresponse up to 598.6 A W^−1^, 1613.8, 6.7 × 10^14^ Jones and 0.88 μs, respectively.

### 4.2. Imaging Applications

The commonly used imaging system is shown in Figure 7. The imaging object is fixed on the X–Y scanning platform between the laser and the perovskite photodetector. The X–Y moving platform is controlled by computer software programming, and the object can move in the horizontal and vertical directions to the X–Y direction. The photodetector is connected to the phase-locked amplifier. By moving the object, the current signal matching the object position and the corresponding position coordinates can be collected. The measured dark current value represents the background noise level of each pixel, and the photocurrent value represents the light intensity level. Finally, the data is transferred to the computer and converted into gray values to realize the image presentation.

As an important optoelectronic device, perovskite UV image sensors have attracted increasing research attention in recent years. Li et al. [69] constructed a UV imaging system using a heterojunction photodetector with an In/GaN–Cs_3_Cu_2_I_5_/Au structure. The abbreviation of the hollow ultraviolet logo ‘UV’ and the solid symbol of a deer were tested under laser irradiation at 325 nm. The imaging results showed that the resolution of the two images was clear and coincided well with the shape of the photomask, demonstrating the stability of this photodetector UV imaging system. In 2020, a UV image sensing system was also prepared by Liang et al. [70], and the perovskite photodetector structure used in the experiments was Cs_3_Cu_2_I_5_/Si core/shell NWs/Au, with a metal mask for the imaging object and the pattern with the letter Z. With this imaging system, it was easy to observe the shape of the letter Z, indicating the high-fidelity property of the perovskite photodetector imaging system based on the core–shell structure of Cs_3_Cu_2_I_5_ NWs.

Compared with the above perovskite photodetectors, Gu et al. [71] prepared a 1024 pixels image sensor based on a perovskite photodiode of MAPbI_3_ NWs 3D array and verified its imaging function by identifying various optical modes projected on the sensor. It did not need the imaging system to image, because the image sensor had high pixel density, a large active area and fast response. Each NW can be considered as one pixel, and this particular design enabled the device to be of extremely high resolution, close to the optical diffraction limit. MAPbI_3_ NWs were sandwiched between 32 top ITO finger electrodes and 32 bottom gold finger electrodes, which were connected to rows and columns using two separate 32-to-1 multiplexes. This allowed the processing of specific pixels by selecting the corresponding row and column numbers. Through the experiment of images and videos, it was found that this perovskite image sensor could capture both static images and dynamic videos of the movement of light spots along a square path. Furthermore, the device could also be applied on flexible substrates to fabricate flexible perovskite optical sensor arrays; this is expected to find applications in wearable electronics, electronic eyes, versatile robots, artificial skin, and so on.

Li et al. [72] developed a perovskite photodiode based on small molecule CO_i_8DDFIC and MAPbBr_3_ heterojunction structures for electrically tunable single/double-color imaging, which could convert the double-color image obtained after imaging through an imaging system to single-color image by applying a small voltage. The photodetector had a cut-off wavelength of 544 nm in the visible region and 920 nm in the NIR region. To investigate the imaging capability of this single/double-color photodetector, an open letter ‘N’ was fabricated as an object on a black acrylic plate. After the action of the imaging system, the ‘N’ images had clear boundaries in both visible and NIR light regions. With a bias of 0.6 V applied to the detector, the dual color imaging was transformed into monochromatic imaging, and the letter ‘N’ image of the NIR light region disappeared. This work has important implications for the imaging of photodetectors in complex environments.

In 2021, Wu et al. [73] fabricated CsPbBr_3_ thin films with precise pixel position, controllable morphology and uniform size using vacuum-assisted drip technology, with an active layer of 2.4 µm ultrathin 10 × 10 perovskite photodetector array. Employing a water-resistant Parylene-C film as the substrate and encapsulating layer, the perovskite film was protected from penetration by polar solutions and effectively isolated air and water. Thus, this detector array exhibited long-term stability and good mechanical stability. The device was a Parylene-C/Au interdigital electrode/SiO_2_/CsPbBr_3_/Parylene-C MSM configuration. The detector array had an ultralight weight (3.12 g m^−2^), was 25 times lighter than standard A4 paper (80 g m^−2^), and could attach well to irregular 3D object surfaces. Notably, the perovskite photodetector arrays with a hemispherical planar support could mimic the structure of the human eye well to prepare an artificial retina for perceptual recognition in imaging applications. 

### 4.3. Biological Detection

Perovskite photodetectors can be combined with the human body to detect human health conditions, for example, in wearable electronic devices worn on the surface of human organs, especially the skin surface. Because the tissue penetration depth of photons in the spectral range NIR-I (700–1000 nm) and NIR-II (1000–1800 nm) can reach 1–20 mm with NIR, photodetectors responsive to biomolecules and fluorescent information on tissue structure can be captured [74]. Figure 8 is the schematic diagram of the photoelectric heart pulse sensor. 

Bao et al. [75] integrated the generation and reception of light signals into a device that allowed bidirectional light signal transmission between two identical perovskite photodiodes. The device structure of the perovskite photodiode was ITO/polyethyleneimine ethoxylated (PEIE)/ZnO/FAI/TFB/MoO_x_/Au, which can be switched between emission and detection modes by changing the external bias, that is, between a perovskite light emitting diode (LED) and a perovskite photodetector mode. A photo cardiac pulse sensor was constructed by preparing two perovskite photodiodes on one chip. The operating principle of the device was as follows: the first perovskite photodiode operated in the LED mode, which transmitted NIR light from the LED to the finger. Part of the light was absorbed by the cutaneous vascular bed, and part of the light was transmitted and reflected by tissues. Another photodetector worked as a photodetector by receiving the arterial pulsation light pulse signal as the heart periodically contracted and relaxed, thus collecting a cardiac pulse signal. The measured cardiac cycle was compared with that measured by commercial pulse sensors, including inorganic green LEDs and silicon photodiodes, whose cycles coincided.

Perovskite photodetectors can combine with a noninvasive glucometer, utilizing the Lambert–Beer law *A = log(I*_0_*/I_t_) = log1/T = εcd*. *A* is absorbance, *I*_0_ is incident light intensity, *I_t_* is transmitted light intensity, *T* is transmittance, *ε* is the molar extinction coefficient, *c* is the concentration of the solution being tested, and *d* is the optical path or thickness of the absorption medium. Thus, in *ε*, under the premise of determination, when *d* is fixed, the the concentration *c* can be obtained according to the absorbance *A*, and the concentration change in human blood glucose can be measured, which immediately reflects the condition of people with diabetes so that subsequent targeted treatment can be undertaken. Noninvasive testing, compared with invasive testing—which does not use venous blood collection in puncture patients—can reduce the physical and mental distress of people with glycosuria. The ability to observe whether a patient’s blood glucose situation is different or not has facilitated the development of biological medical devices by visualizing the curve of the magnitude of blood glucose change during a day. 

### 4.4. Optical Communication and Up-Conversion Systems

Photodetectors are key components in optical communication systems that realize signal-efficient photoelectron conversion, and their response speed directly determines the bandwidth of the whole system [76]. Therefore, they can be applied to optical communication systems. The light communication system consists of an emission system, a modulation module, a driver circuit, a light source, a demodulation module, a driver circuit and a receiving system. As shown in Figure 9, the working principle is as follows: the driver converts the data flow of the computer to high and low levels, and the voltage levels drive the light source, producing modulated light. A perovskite photodetector receives modulated light and generates modulated photocurrents, which are input into the driver at the receiving end. The driver converts the voltage signal into data, which is transmitted to the computer.

In 2018, Bao et al. [76] fabricated a high-performance p-i-n type photodiode based on all inorganic CsPbI_x_Br_3−x_ perovskite film, and the detector substrate was PEIE. The device structure was ITO/PEIE/PTAA/CsPbI_x_Br_3−x_ /PCBM/BCP/Ag. Here, the CsPbI_x_Br_3−x_ perovskite layer was composed of CsPbIBr_2_ and CsPbBr_3_, respectively. Since the detection limits of CsPbIBr_2_ and CsPbBr_3_-based photodetectors were tens of pW cm^−2^, the response was fast, and all inorganic perovskites exhibited excellent environmental stability. Bao et al. first applied CsPbIBr_2_ photodetectors as optical signal receivers in a visible light communication (VLC) system to detect its optical communication ability. The experiment showed that this VLC system can receive modulated photocurrent signals at different bit rates, the expected maximum transmission rate could exceed 1 Mbps and the computer at the receiving end could accurately receive text and audio information emitted from another computer. Thereafter, Bao et al. [75] fabricated two identical perovskite photodiodes whose structures were ITO/PEIE/ZnO/FAI/TFB/MoO_x_/Au. This photodiode could be switched between emission and detection modes by changing the external bias. It was found by testing that the PL spectra and absorption spectra of the photodiode perovskite film layer exhibited a large overlap with only a 40 meV Stokes shift, indicating that the perovskite film had similar light sensitivity to the same film emission. Based on the two functions of this photodiode, they can be integrated into an optical communication system, detecting the bidirectional optical communication ability of the system.

Through the experimental verification with the dual-function perovskite diode optical communication system, it was found that the system could achieve bidirectional communication, receiving text and audio information. This indicated that the application of perovskite photodiode in optical communication was feasible. Although the speed of this perovskite photodiode is expected to reach hundreds of Hz, even gigahertz, the values are much lower than that of a commercial inorganic semiconductor device, which also has a lower bandwidth than that required for fiber and visible light communication systems. However, it showed great potential in medium and low-speed optical communication, as well as inter-chip and intra-chip data links for optical integrated circuits. The bifunctional perovskite photodiodes not only simplified the design and size of optical communication systems, but also facilitated the development and commercialization of different functional perovskite photodetectors.

Zhao et al. [77] prepared an infrared up-conversion system using a photodetector with the structure of ITO/PTAA/phenylethylammonium iodide (PEAI)/Cs_0.05_MA_0.45_FA_0.5_Pb_0.5_Sn_0.5_I_3_/PEAI/C60/BCP/Cu, and the spectral range detectable by the detector was 300–1050 nm. This infrared up-conversion system mainly consisted of an amplified circuit composed of LM324, a metal oxide semiconductor tube IRF530N and a white LED. When the photodetector received a NIR light signal, it converted the optical signal into an electrical signal. After being amplified by the amplification circuit, the IRF530N turned on the switch, and the white LED light was illuminated. The white LED light emission indicated that the Sn-Pb perovskite photodetector could convert the NIR signal into a visible light signal, which was of great significance to the visualization of NIR light.

### 4.5. Polarized Light Imaging

In recent years, polarized light and structured light have attracted much attention of researchers [78]. Perovskites are sensitive to polarized light and can realize polarization imaging. Polarized light imaging can obtain multi-dimensional polarization information in a complex background. It can be used to detect cancer cells and tissues in the medical field, as well as weak optical detection. Song et al. [79] made an efficient moiré MAPbI_3_-laminated double shallow grating photodetector on the digital multi-function disc (DVD) through the imprint lithography process. It localized the photons and achieved sensitive digital polarization imaging. The light responsivity of the photodetector was 15.62 A W^−1^ and the specific detectivity was 5.58 × 10^13^ Jones. Zhang et al. [80] prepared a polarized light-sensitive photodetector using a 2D/3D ((4-AMP) (MA) _2_Pb_3_Br_10_/MAPbBr_3_) perovskite SC heterostructure. The novelty of the photodetector was that it combined the anisotropy of 2D perovskite materials with the high carrier transmission capacity of 3D perovskite. In addition, due to the strong built-in electric field generated at the interface of 2D/3D heterogeneous crystals, the device could effectively separate photogenerated carriers and inhibit carrier recombination under zero-bias operation [80].

Circularly polarized light is widely used in 3D display [81], quantum computing [82], and chiral sensors [83]. Chiral cations are introduced into perovskite quantum wells to endow 2D perovskite with intrinsic chirality, making it possible to detect perovskite circularly polarized light. Wang et al. [84] synthesized (R)-β-MPA]_2_MAPb_2_I_7_ SC and chiral quasi-2D perovskite thin film by introducing chiral β-methylphenethylamine into the structure, wherein ((R)-β-MPA = (R)-(+)-β-methylphenethylamine). Based on this, Wang et al. constructed a PET/[(R)-β-MPA]_2_MAPb_2_I_7_/Au photodetector. The chiral quasi-2D perovskite film had excellent photoelectric properties and strong circular dichroism. The flexible detector had a high responsivity of up to 1.1A W^−1^ and 2.3 × 10^11^ Jones detectivity. After 100 cycles of bending, the photocurrent and anisotropy factors of the detector were less than 10%, showing that the flexible film chiral quasi-2D perovskite photodetector had excellent stability.

## 5. Improvement and Optimization

### 5.1. Lead-Free Materials

Lead-based perovskite has become a ‘star’ in the perovskite industry because of its strong light absorption ability, high carrier mobility, strong photoelectric conversion ability, high defect tolerance and strong stability. However, as we know, lead is a kind of heavy metal that is very harmful to the human body and the environment. It can enter the human body through air, food, drinking water, and in other ways. It can also cause damage to the nervous system, kidneys and cardiovascular system, so as to endanger life. In addition, lead will pollute the environment, affect the growth and development of plants, hinder the photosynthesis of plants and pollute the atmospheric environment. Therefore, it is very necessary to seek new elements to replace lead in order to reduce the harm of lead-based perovskite. At present, there are four common methods to replace lead in lead-based perovskite. Table 3 summarizes and compares the optical detection performance of various lead-free perovskite photodetectors.

The first method is to replace the elements belonging to the IVA main group with Pb elements, such as Sn and Ge elements, which can reduce the band gap of perovskite. However, Sn^2+^ and Ge^2+^ are easily oxidized to stable valence Sn^4+^ and Ge^4+^ in the air, which will bring defects to the perovskite layer, lead to its phase transformation or decomposition and affect the performance and stability of the photodetector. In 2020, the Shen Liang team of Jilin University used the Sn element to replace Pb, widened the spectral range to 1050 nm, realized NIR light response, and creatively put forward the double-sided passivation strategy. In addition, they wrapped the bottom and top of Sn-Pb perovskite film with PEAI, effectively isolated the damage of water and oxygen to perovskite layer, and improved the stability of Sn^2+^ in the atmospheric environment [76]. Furthermore, the 3D–2D mixed layer was used in the perovskite layer. The 2D perovskite layer also made the photodetector more stable. At present, the stability and photoelectric performance improvement of Sn^2+^ and Ge^2+^-based perovskite photodetectors still need further research and exploration.

The second method is to use the element Bi of the VA group to replace Pb. The outermost electron arrangement of the Pb element is 6s^2^6p^2^. Ions with a 6s^2^6p^0^ electronic configuration can be found to form an electronic structure similar to Pb-based perovskite. Therefore, the cope iodic elements thallium (6s^2^6p^1^) and bismuth (6s^2^6p^3^) are taken into account. For example, Tl^+^ and Bi^3+^ ions. However, thallium is highly toxic, and the toxicity of Tl^+^ is much stronger than that of Pb^2+^, so only the replacement of Bi^3+^ ions, such as Cs_3_BiBr_6_, Cs_3_Bi_2_I_9_, Cs_3_Bi_2_Br_9_, CsBiI_3_, double perovskite Cs_2_AgBiBr_6_, Cs_2_AgBiCl_6_ and other Bi containing perovskites, is considered. In 2021, Li et al. [88] integrated Cs_3_Bi_2_I_9_ SC film on a Si wafer. The experiment found that the n-Si (111) plane and the p-Cs_3_Bi_2_I_9_ (001) plane had good lattice matching and energy band alignment, and the lattice mismatch was only 0.52%, indicating that Cs_3_Bi_2_I_9_ SC film could grow well on Si crystal plane with few grain boundaries and defects. The photodetector had high sensitivity, a switching ratio of 3000, and fast light response time of 1.5 μs, which was higher than most lead-free halide perovskite photodetectors at present. In the same year, Liu, Gao, and others [89] prepared self-powered UV photodetectors using NiO_x_/Cs_3_Bi_2_Br_9_ heterojunction structure. Nickel oxide was deposited by electrochemical deposition as an electron transport layer to realize UV imaging. The optical response time was 3.04/4.65 ms. LDR was 116.6dB. When the bias voltage was 0 V, the responsivity was 4.33 mA W^−1^ and the specific detectivity was 1.3 × 10^11^ Jones.

The third method is to use +2 valence transition metal ions to replace Pb^2+^, mainly including Zn^2+^, Mn^2+^, Fe^2+^, and Cu^2+^. Due to their small size, the 3D perovskite structures will become a layered structures, and the current research results of Cu^2+^ ions, such as CsCu_2_I_3_ [38], Cs_3_Cu_2_I_5_ [24] are the best [91,92,93]. In addition, tellurium (Te) can be used to replace Pb, that is, Te^4+^ ions. Xu et al. used a stable compound of iodide system AX-TeX_4_ (where A is Rb, C_S_ or Tl, and X is Br or I) with general formula A_2_TeX_6_ [94,95], to prepare 0D perovskite Cs_2_TeI_6_. Cs_2_TeI_6_ was composed of high atomic number elements with high resistance and high air and moisture stability, which made it suitable as a sensitive X-ray photoconductor and for use in manufacturing X-ray detectors [96].

The fourth method is to use antimony (Sb) instead of Pb. Yang et al. [85] synthesized a low trap states density of 2.9 × 10^10^ cm^−3^, a high carrier mobility of 12.8 cm^2^ V^−1^ S^−1^, and a carrier diffusion length of 3 μm MA_3_Sb_2_I_9_ SC. The photodetector structure was ITO/MA_3_Sb_2_I_9_/ITO. Under the irradiation of 460 nm monochromatic light, the response could reach 40 A W^−1^ and the response speed was less than 1 ms. The performance of the detector was comparable to that of most lead-based perovskite photodetectors. When the device was placed in the surrounding environment for more than two weeks, the optical response of the detector changed little and could be stored stably.

### 5.2. Stability

As we know, compared with silicon and other highly stable inorganic semiconductors, perovskite has insufficient stability and cannot be stored for a long time. The material is easy to decompose and phase change under the influence of external temperature and humidity, which has a great impact on the performance of optoelectronic devices. In particular, due to the existence of organic components, the stability of organic–inorganic hybrid perovskite is greatly reduced compared with all inorganic perovskite, and it is easy to degrade into a hydrated phase when exposed to humid air [97,98,99]. As perovskite is easy to decompose, it can only be stored in the surrounding environment for a short time. For example, MAPbI_3_ is easy to decompose into MAI and PbI_2_ when exposed to high humidity, strong light, oxygen enrichment and high temperature. In addition, FAPbI_3_ is easy to decompose into FAI and PbI_2_. However, FAPbI_3_ is very sensitive to light. α-FAPbI_3_ (photosensitive phase) has good photoelectric performance. Unfortunately, α-FAPbI_3_ is unstable, because it is too large for the perovskite cage, which can easily be converted into β-FAPbI_3_ and the yellow phase (δ-FAPbI_3_) [100]. The following reaction equation explains the chemical change of MAPbI_3_ decomposition [101].
CH_3_NH_3_PbI_3_(s) ⇌ CH_3_NH_3_I(aq) + PbI_2_(s)(1)
CH_3_NH_3_I(aq) ⇌ CH_3_NH_2_(aq) + HI(aq)(2)
4HI(aq) + O_2_ ⇌ 2I_2_(s) + 2H_2_O(3)
2HI(aq)→H_2_↑ + I_2_(s)(4)

For FAPbI_3_, the common method to improve the stability of FAPbI_3_ is to blend Cs^+^, MA^+^, Ru^+^, and Br^−^ into FAPbI_3_, partially replacing the large radius cations at position A and halogen anions at position X, respectively [102,103]. For MAPbI_3_, when Cl and Br are doped into MAPbI_3_, the humidity stability increases significantly. This multi-ion mixture perovskite is relatively stable and can inhibit ion migration, which has been widely studied in recent years. In addition, perovskite SCs has been widely used in the preparation of perovskite photodetectors due to their low trap state density, lack of grain boundary, longer carrier diffusion length (~10 μm) [104,105], higher carrier mobility (~100 cm^2^) [106], high photon absorption and relatively strong stability. Previously, Chen et al. [103] mixed Cs^+^ and MA^+^ into FAPbI_3_, and found that the content of 5% Cs, 10% Br, and 5% MA could maintain the stability of perovskite structure. An appropriate amount of Br ion could change the lattice contraction, release lattice stress, inhibit potential phase separation tendency, and balance the radius difference between FA and Cs such as triple-cation mixed-halide (FAPbI_3_) _0.9_ (MAPbBr_3_) _0.05_ (CsPbBr_3_) _0.05_ SC. Experiments showed that it not only had good stability (10000 h water oxygen stability, 1000 h light stability, and excellent thermal stability); moreover, it had a narrow band gap (1.52 eV) which was very suitable for preparing perovskite solar cells and perovskite photodetectors. Liu et al. [67] doped Cs^+^, MA^+^, and Br^−^ in FAPbI_3_ and added FAH into the perovskite precursor to prepare pure phase, high-quality FA_0.9_Cs_0.05_MA_0.05_PbI_2.7_Br_0.3_ SCs; wherein, FAH effectively inhibited the oxidation of I in FAPbI_3_ to I_2_ and I_3_^−^, and the deprotonation of FAH^+^ and MAH^+^, which hindered the formation of δ-FAPbI_3_ and δ-CsPbI_3_.

For the problem of improving the stability of perovskite, an effective method is to improve the stability by controlling the composition and optimizing the Goldschmidt tolerance factor [107,108]. The Goldschmidt tolerance factor was proposed by Goldschmidt in 1926 to describe the relationship between material stability and ion size [109]. The calculation formula of tolerance factor is *t = (r_A_ + r_B_)/[2(r_B_ + r_X_)]*^1/2^, where *r_A_* is the size of the cation at position A, *r_B_* is the size of metal ions at position B, *r_X_* is the size of X-position halogen ion. When 0.9 < *t* < 1, perovskite is an ideal cubic phase [28].

Another method is to inhibit ion migration in perovskite by passivation. Defects generally exist on the surface or grain boundary of perovskites, and electron–phonon coupling leads to the formation of trap states in perovskite. The perovskite surface can be passivated to remove surface defects, or the grain boundary can be passivated to induce the perovskite to produce large grains, thus producing a dense and uniform film. Metal ions are usually doped into perovskite to passivate defects by interacting with ion bonds between organic cations and negative charge defects of perovskite. Common metal ions include Na^+^, K^+^, Cu^+^, Ag^+^, Rb^+^, etc. [27,28,110,111]. Bi et al. reported for the first time that the metal ion Na^+^ had a passivation effect, and explained that the negatively charged defect of Na^+^ passivation was either through the MA vacancy adsorbed on the grain boundary, because it had a similar size to and same valence charge as MA^+^, or formed through an ionic bond with an uncoordinated halide [27]. In addition to metal ions, other ions could also be used for passivation. For example, NH_3_^+^ passivated negative charge defects through electrostatic interactions, including ionic bonds and hydrogen bonds [112]. Ions with ammonium functional groups such as BA^+^, PEA^+^, and OA^+^ have been proven to improve the stability of perovskite [48,77,113,114,115]. An OA^+^ decorative surface has also been proven to improve the thermal stability and moisture stability of organic–inorganic hybrid perovskite materials [115]. Wu et al. [116] added the ionic liquid 1-butyl-3-methylimidazolium tetrafluoroborate (BMIMBF4) as an additive to MAPbI_3_ NWs, which effectively passivated the defects, inhibited the decomposition of MAPbI_3_ NWs, and improved performance and stability. The experiment showed that the unpacked MAPbI_3_ nanowire detector could still maintain the initial performance of 100% when exposed to the surrounding environment for more than 5000 h.

Furthermore, the interfaces of different layers of the perovskite photodetector can be treated. For example, PCBM/C60 is coated on the perovskite as an electron transport layer, which passivates the charge defects. C60 can also be used as a hole barrier layer to effectively prevent the reverse injection of holes, reducing the dark current and device noise [117]. For photodiodes, the selections of electron transport layer and hole transport layer are also very important for the overall stability of the detector. The traditional hole transport layer generally uses spiro-OMeTAD. Researchers usually add lithium bis trifluoromethane sulfonamide (Li TFSI) and 4-tert-butylpyridine (TBP) to it; however, lithium salt has strong water absorption, which will cause pinholes and voids in the perovskite film. Zhang et al. [118] added RbI together with Li TFSI and TBP into the spiro-OMeTAD. Because of the complexation between RbI and TBP, the evaporation of TBP was prevented and the aggregation of Li TFSI was hindered, thus reducing the generation of defects in perovskite MAPbI_3_ films. Therefore, compared with the spiro-OMeTAD, RbI doped spiro-OMeTAD not only improved the conductivity of the hole transport layer and promoted the energy level matching with the perovskite layer, but also improved the stability of the device.

## 6. Prospect

As shown in Figure 10, photodetectors are widely used in X-ray detection, wearable devices, intelligent vehicles, artificial intelligence, imaging, communication and other fields. The following seven points are the outlook for the future development of perovskite photodetectors.

(1) With the rapid development of science and technology, the arrival of 5G intelligence and the information explosion era, optical signals now carry a large amount of information. The application of perovskite photodetectors in optical signal transiting, converting, storing and processing has become more attractive. Thus, improving the performance and stability of perovskite photodetectors, combined with the ‘Internet of Things’, can accelerate the development of digitalization, intelligence and informatization in modern society.

(2) Halide perovskite, as a new type of X-ray direct detection material, has a strong X-ray absorption capacity and excellent carrier transmission capacity. Moreover, it can be prepared at a low temperature and low cost, is easy to combine with polysilicon arrays, and is safer for human body in particular. In addition, perovskite photodetectors can be combined into imaging systems to improve the imaging resolution. Thus, they are expected to play a major role in the medical field in the future.

(3) Compared with polycrystalline thin films, SCs have better stability and carrier transmission capacity, and fewer grain boundaries and defects. However, the reported perovskite SC photodetectors featured no significant performance improvement when compared with polycrystalline products. Therefore, strategies including improving the preparation process of perovskite SCs and growing flexible, large-area ultra-thin perovskite SC films are desirable for SCs devices [119,120].

(4) Due to the large specific surface area of 2D materials, they can produce a super-strong grating pressure effect. This grating pressure effect further increases the carrier transport capacity of channel materials, thus obtaining an extremely high optical gain effect [45,121,122]. Therefore, the combination of perovskites with high carrier mobility and high spatial resolution imaging materials (such as graphene, 2D transition metal sulfides and other semiconductors) to form a heterojunction structure is a trend that will factor in the preparation of perovskite photodetectors in the future. In addition, research on low-dimensional perovskites such as the preparation of ordered 1D NWs and nanorod arrays can improve the responsivity and optical detection ability of perovskite photodetectors and reduce dark current. In the future, research may focus on the preparation process and methods of 2D perovskite photodetectors to improve the thermal, optical, water and oxygen stability. Lead-free materials may be used to develop new 2D perovskite materials, and deeply investigate the working mechanism, interface carrier transport, device performance optimization, etc., of 2D perovskites. In addition, because the doping polarity of perovskite QDs is opposite to that of most 2D materials, they can be combined with 2D materials to adsorb long-lived charges as electron capture sites, then realize large built-in electric fields and promote efficient carrier separation and vertical injection channels.

(5) These photodetectors can be used in the biological field for bioluminescence detection. They can be combined with a non-invasive blood glucose meter to detect the glucose concentration of the human body with NIR light, so as to reduce the physical and mental pain of patients when measuring blood glucose [72,123,124]. Ultraviolet perovskite photodetectors can be used for flame monitoring, combustion controlling, solar radiation measurement, ultraviolet disinfection and ultraviolet communication.

(6) Perovskite photodetectors can be combined with imaging systems to realize face recognition in the field of computer communication. Integrated within a business model, they can realize the industrialization and application of detectors. They can be combined with solar cells to realize self-driven perovskite photodetection, and carry out energy conversion and photoelectric detection simultaneously.

(7) Chiral perovskite photodetectors are well employed to realize efficient circularly polarized optical detection. This strategy avoids using complex optical systems prepared by traditional physical methods, which consist of linear polarizers, quarter glass slides and traditional non-chiral photodetectors. The study on flexible and micro devices is a promising area in the future.

## 7. Conclusions

In short, whether in the selection of materials for each layer, the combination of different materials, the device structural innovation, the performance improvement of devices or the combined application of different fields, perovskite photodetectors still have great development potential, and there is a long way to go to their commercialization. We believe that in the future, through the joint efforts and exploration of different scientists and researchers in various fields, perovskite photodetectors can be applied to more aspects of human life and boost greater potentiality.

## Figures and Tables

**Figure 1 nanomaterials-12-04390-f001:**
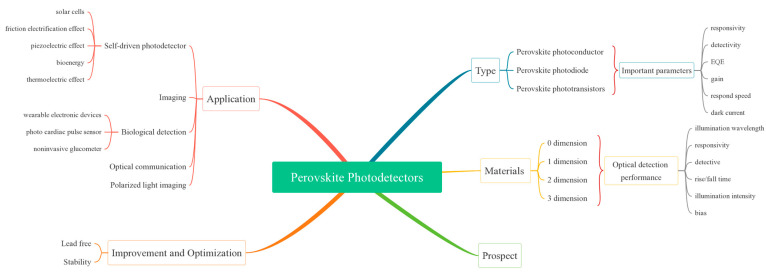
The roadmap of current work.

**Figure 2 nanomaterials-12-04390-f002:**
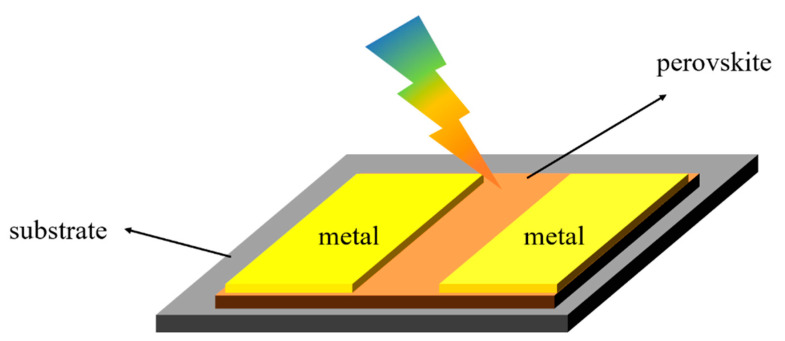
Structure diagram of the perovskite photoconductor.

**Figure 3 nanomaterials-12-04390-f003:**
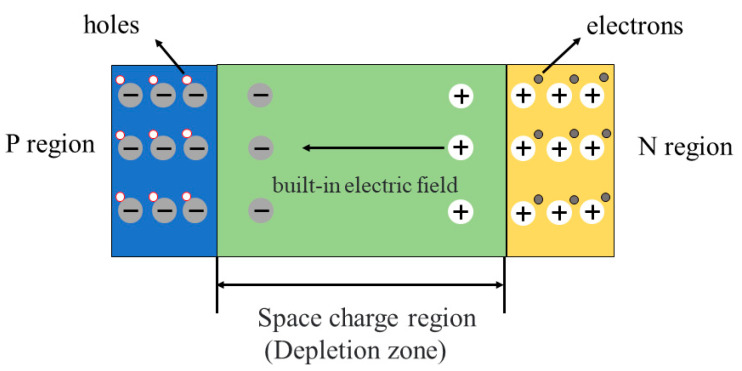
Schematic diagram of p-i-n junction.

**Figure 4 nanomaterials-12-04390-f004:**
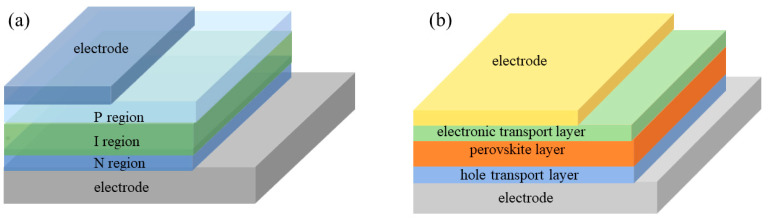
Structure diagram of (**a**) the p-i-n type photodiode and (**b**) the p-i-n type perovskite photodiode.

**Figure 5 nanomaterials-12-04390-f005:**
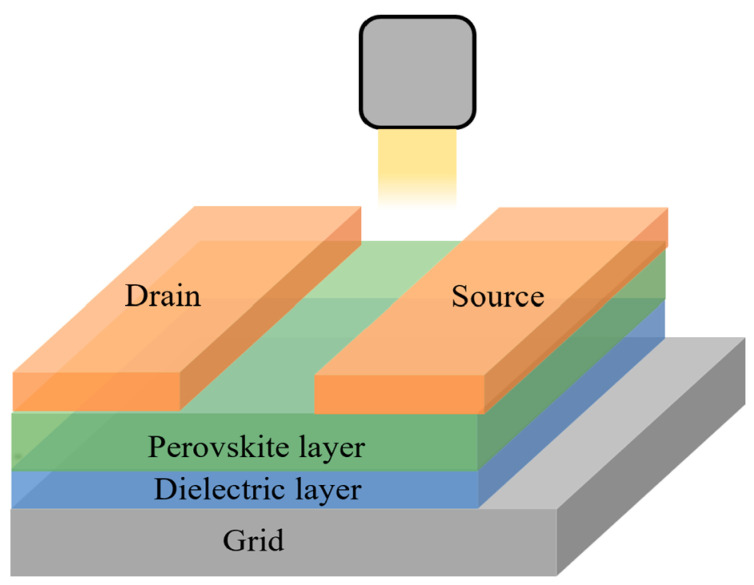
Structure diagram of the perovskite phototransistor.

**Figure 6 nanomaterials-12-04390-f006:**
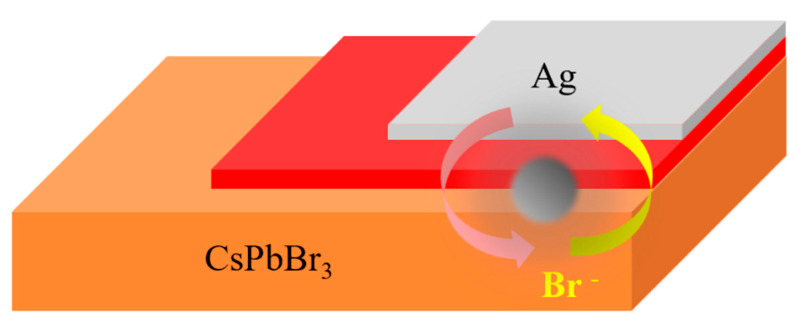
Br^−^ ions migrate into the Ag electrode to produce AgBr precipitation.

**Figure 7 nanomaterials-12-04390-f007:**
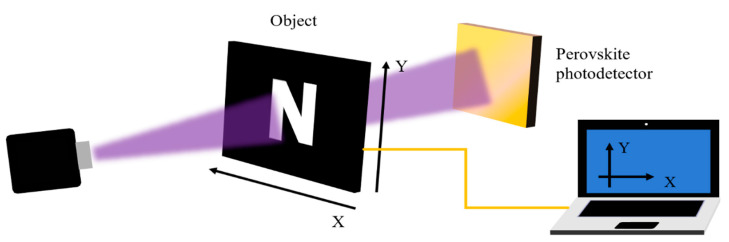
Schematic diagram of the imaging system.

**Figure 8 nanomaterials-12-04390-f008:**
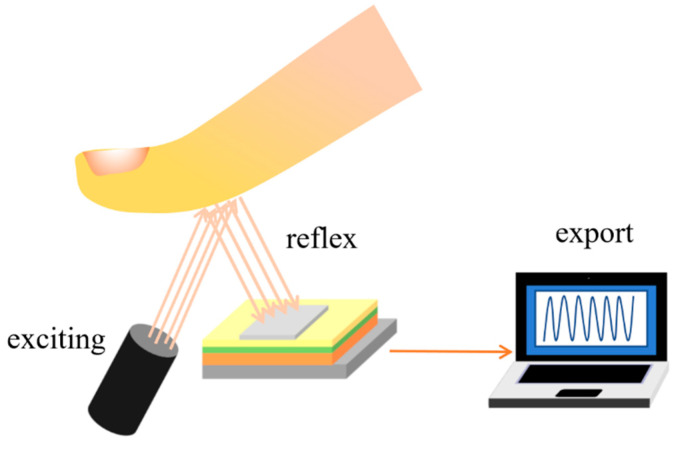
Schematic diagram of the photoelectric heart pulse sensor.

**Figure 9 nanomaterials-12-04390-f009:**
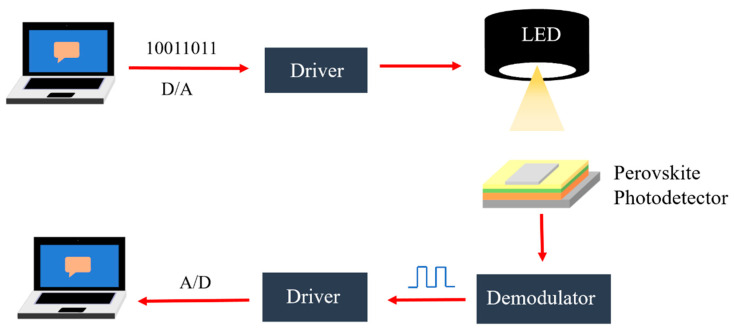
Schematic diagram of optical communication system.

**Figure 10 nanomaterials-12-04390-f010:**
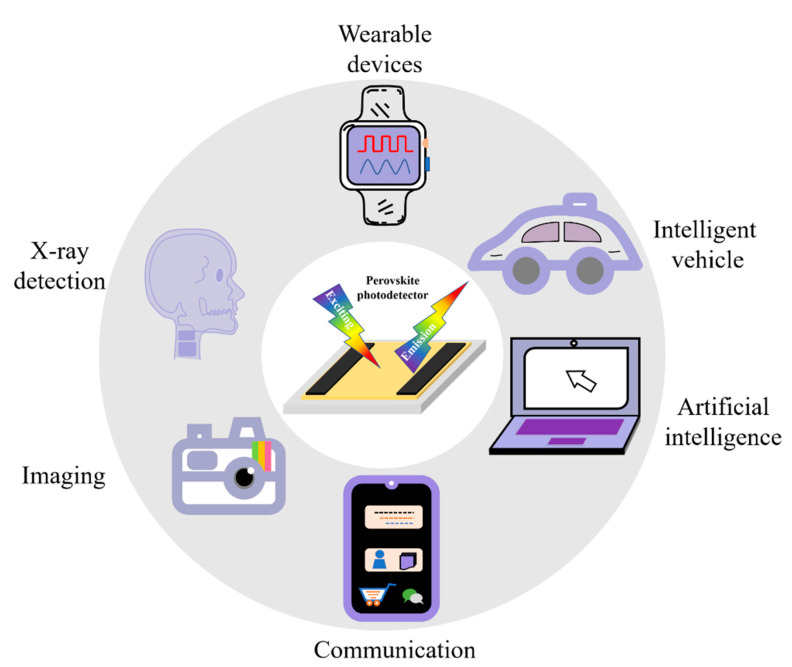
Perovskite photodetectors applications.

**Table 1 nanomaterials-12-04390-t001:** Characteristics of performance parameters of perovskite photoconductors, photodiodes, and phototransistors.

Performance Parameter	Perovskite Photoconductor	Perovskite Photodiode	Perovskite Phototransistor
Responsivity	high	low	high
Detectivity	low	high	high
EQE	high	low	high
Gain	high	low	high
Respond speed	slow	fast	slow
Dark current	high	low	high

**Table 2 nanomaterials-12-04390-t002:** Performances of low-dimensional perovskite photodetectors based on the above literature.

Device Structure	Dimension	Illumination Wavelength (nm)	Responsivity (A W^−1^)	Detectiviy (Jones)	Rise/Fall Time (ms)	Illumination Intensity (mW cm^−2^)	Bias (V)	Year (Reference)
ITO/Cs_3_BiBr_6_/ITO	0D	400 nm	0.025	0.8 × 10^9^	50/60	4	6	2019 [23]
ITO/Cs_3_Cu_2_I_5_/ITO	0D	265 nm	0.0649	6.9 × 10^11^	26.2/49.9	/	1	2019 [24]
Si/Au/ (DME)PbBr_4_/Au	1D	375 nm	132.3	/	105/117	0.001	20	2017 [35]
Si/CsCu_2_I_3_/Au	1D	325 nm	32.3	1.89 × 10^12^	0.00694/0.214	/	5	2020 [38]
quartz substrate/Ca_2_Nb_3_O_10_/Cr/Au	2D	280 nm	14.94	8.7 × 10^13^	0.08/5.6	/	3	2021 [49]
ITO/(C_4_H_9_NH_3_)_2_PbI_4_/(C_4_H_9_NH_3_)_2_(CH_3_NH_3_)Pb_2_I_7_/Cr/Au	2D	540 nm	0.69	/	150/170	45	3	2019 [50]
ITO/(BA)_2_(MA) Pb_2_I_7_/Au	2D	620 nm	/	1 × 10^11^	125/74	0.02	5	2019 [51]

**Table 3 nanomaterials-12-04390-t003:** Comparison of optical detection performance of lead-free perovskite photodetectors reported in the previous literature.

Device structure	Material Structure	EQE (%)	Responsivity (A W^−1^)	Detectiviy (Jones)	Rise/Fall Time (ms)	Spectrum Range (nm)	Bias (V)	Year (Reference)
ITO/Cs_3_BiBr_6_/ITO	Single crystal	0.008	0.025@400 nm	0.8 × 10^9^@400 nm	50/60	300–485	6	2019 [23]
ITO/MA_3_Sb_2_I_9_/ITO	Single-crystalline film	/	40@460 nm	10^12^@460 nm	0.4/0.9	400–600	5	2018 [85]
Si/CsBi_3_I_10_/Au	Polycrystal-ine film	75.2	0.492@808 nm	1.38 × 10^11^@808 nm	0.073/0.036	400–1200	−1	2019 [86]
ITO/Cs_3_Cu_2_I_5_/ITO	Polycrystal-line film	0.3	0.0649@265 nm	6.9 × 10^11^@265nm	26.2/49.9	200–405	1	2019 [24]
FTO//TiO_2_/Cs_3_BiBr_9_/Au	Polycrystal-line film	/	0.006@405 nm	3.39 × 10^11^@405 nm	0.57/0.58	325–500	/	2020 [87]
Si/CsCu_2_I_3_/Au	Nanowire	/	32.3@325 nm	1.89 × 10^12^@325 nm	0.00694/0.214	230–350	5	2020 [38]
Ag/Si/Cs_3_Bi_2_I_9_/Au	Single-crystalline film	/	/	3.9 × 10^11^@450 nm	0.0015/0.422	250–500	3	2021 [88]
FTO/NiO_x_/Cs_3_Bi_2_Br_9_/Ag	Polycrystal-ine film	/	0.00433@450 nm	1.3 × 10^11^@450 nm	3.04/4.66	300–550	0	2022 [89]
FTO/Au/TiO_2_/Cs_2_AgBiBr_6_/CuSCN/Au	Polycrystal-ine film	/	0.34	1.03 × 10^13^@405 nm	28.75/32.95	/	0	2021 [90]

## Data Availability

Not applicable.

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
