# Peer review of "A Review of Perovskite-Based Photodetectors and Their Applications"

_nanomaterials, 2022, doi:10.3390/nano12244390_

Round 1
Reviewer 1 Report
Comments to
The process of perovskite materials in photodetectors, by Hai yan Wang , Yu Sun , Jin Chen * , Feng chao Wang * , Rui yi Han , Can yun Zhang , Jin fang Kong , Lan Li , Jing Yang
The manuscript focuses on an interesting topic, due to the attractive properties of perovskite-based materials for optoelectronic devices, in particular for photodetectors. However, having in mind that several review articles have been published on this topic in recent years, my first comment is that the authors must highlight what are the relevant contributions of this work. In order to highlight the novel contributions of the present manuscript, it would be necessary to comment on the central points (and, where appropriate, the aspects not included) in recent review articles.
Considering the content of the article, I think that the title is not informative. What do the authors mean by 'process' in the title?
On the other hand, the way of reporting the characteristics of the devices that are referred to in the manuscript should be improved. Grammar structures such as:
" the intensity was 18.5 μW cm-2 under the irradiation of incident light…" (in page 6, lines 245-246 ) or
“Under the irradiation of 620 nm wavelength light with 5 V bias voltage, the light amplitude illumination was 20 μW cm-2…” (in page 9, lines 265-266) or
“…synthesized low trap density of states 2.9×1010 cm-3…“ (in page 16, lines 487-488)
and some others sound strange to me. In addition to revising the wording of some sentences, the inclusion of comparative tables would result in greater clarity.
Along with this, I have a number of comments and criticisms about the manuscript:
1. Beginning with some general issues, in my opinion there are formal aspects that must be corrected; specifically, there are sentences that are not clear, as well as other sentences that I consider unnecessary; in addition, several graphic schemes seem too simple for a scientific review.
For example, Figures 6, 9 or 11 do not provide relevant information for the readers of a scientific article. The authors say that Figure 2 is the “…schematic diagram of the p-i-n perovskite”, but is the basic scheme of any p-i-n structure.
On the other hand, the drawings of some devices (such us phototransistor in Figure 5) are assigned to "perovskite devices", although they are general schematics.
2. Some general paragraphs, such as those devoted to the fundamentals of some phenomena, should be revised; for example, in page 3, subsection 2.2 Perovskite photodiode, the description of a PIN junction is somewhat repetitive. I suggest a structure beginning with the description (brief) of general characteristics of photodiodes based on PIN junctions and then, in more detail, highlighting the advantages of using perovskite materials.
On the other hand, there are some sentences that are unnecessary (in the context of a scientific article). Por example, in page 8, lines 219 and 220: "… 2D perovskite refers to perovskite materials with 2D geometric structures. In other words, perovskite materials have large planes and thin thickness…"
Similar comments would be applicable to other sections of the manuscript. In my opinion the structure of quite a few paragraphs needs to be revised to make the manuscript easier to read.
3. The repetition of similar sentences or analogous explanations should be avoided. For example, in section 3.1, page 6, the description of 0D perovskites is repeated almost verbatim in:
-Lines 158-160. “0D perovskites are composed of isolated metal halide octahedral anions or metal halide clusters, which are surrounded by organic or inorganic cations with excitons strongly confined to each octahedron [6].”
-Lines 177-179. “…the 0D perovskite has a special structure in which the metal halide octahedra or metal halide clusters are isolated by the surrounding organic or inorganic cations and the excitons are strongly confined by the octahedra”
4. In page 8, line 248 and following, the description of narrowband photodetectors is redundant: “…narrowband photodetectors only respond to a narrow spectrum, collecting photogenerated carriers of specific wavelengths and suppressing photogenerated carriers of other wavelengths […] The principle of its narrowband charge collection mechanism is to realize the narrowband spectral response by only collecting the photogenerated carriers at specific wavelengths and inhibiting the current generated by photogenerated carriers at other wavelengths. [48].
5. Page 12, lines 388 and 389, the definition of absorbance A is wrong (the right expression is A=log(Io/It). On the other hand, I suggest using the word “transmitted” instead of “exited” light intensity.
6. The use of references in the final section (Conclusion and Outlook) is not appropriate.
7. In addition to these aspects, there are a series sentences that need to be clarified/rewritten. Below, I list a few examples that can be read in the first pages, but the manuscript should be revisrd in depth; misprints must be corrected as well.
a) Page 1, in the Introduction, last sentence of the first paragraph. Is the word "As" missing at the beginning of the sentence?
“As for quantum dot photodetectors that have developed rapidly in recent years, although they can be processed in solution…”
The first sentence of the second paragraph should be modified
“With the development of perovskite materials, Due to its high carrier mobility, long carrier diffusion length, adjustable band gap [1-3], and strong light absorption ability [4-6], perovskite materials ...”
In the last sentences of the Introduction section, the verb tenses must be revised; I don't think I understand the meaning of the last sentence:
“We believed that the future perovskite photodetectors would make a qualitative leap in stability, response speed, detectivity, and other aspects. These would be realized industrialization and improved people's life.”
b) Page 7, line 185 and following; It can be read that “Here are three potential research methods…”, but the authors give a list of four methods.
c) There have been quite a few typos in the use of uppercase and lowercase letters throughout the manuscript (I indicate below a few examples, but there are many)
Page 2, line 62: “…MSM absorb incident photons, When the photon energy…”
Page 8, line 244: Van der Waals
Page 12, lines 388 and 389: “it” and “t” should be upper case (“It” and “T” respectively) and “Is” after epsilon must be lower case. (is).
Page 16, line 483: Cs
d) The meaning of the acronyms should be entered the first time the terms are used. For example, Near Infrared is already used in the Introduction, but NIR is not introduced until page 8; PET is mentioned in page 7, "and polyethylene terephthalate (PET)" in page 8.
Considering the aforementioned, my overall assessment is not positive, and I do not recommend the publication of the manuscript in its current state.
Author Response
Response to Reviewers’ Comments
1. The manuscript focuses on an interesting topic, due to the attractive properties of perovskite-based materials for optoelectronic devices, in particular for photodetectors. However, having in mind that several review articles have been published on this topic in recent years, my first comment is that the authors must highlight what are the relevant contributions of this work. In order to highlight the novel contributions of the present manuscript, it would be necessary to comment on the central points (and, where appropriate, the aspects not included) in recent review articles.
Response: Thank you very much for the constructive suggestion. In our manuscript, the work mainly focuses on the perovskite photodetector types, material structure, practical applications (such as imaging, optical communication, biological detection fields etc.), and the avenues of device performance improvement (such as lead-free materials, stability), while that of reported studies (Li et al. Research progress of high-sensitivity perovskite photodetectors: A review of photodetectors: noise, structure, and materials. ACS Applied Electronic Materials, 2022, 4, 1485–1505; Ahmadi et al. A review on organic-inorganic halide perovskite photodetectors: Device engineering and fundamental physics. Advanced Materials, 2017, 29, 1605242; Zhao et al. Recent research process on perovskite photodetectors: A review for photodetector-materials, physics, and applications. Chinese Physics B, 2018, 27, 127806) mainly focuses on device structure design, interface engineering, fabrication approaches, and the physical mechanism of materials and devices. To be clearer and more rigorous, we have state above in the revised manuscript as suggested. In addition, the above-mentioned studies also have been cited in the revised version.
2. Considering the content of the article, I think that the title is not informative. What do the authors mean by 'process' in the title?
Response: Thank reviewer for the kind reminding. To be clear and rigorous, we have revised the title to be “A review of perovskite-based photodetectors and their applications” in the revised manuscript.
3. On the other hand, the way of reporting the characteristics of the devices that are referred to in the manuscript should be improved. Grammar structures such as: “the intensity was 18.5 μW cm-2 under the irradiation of incident light…” (in page 6, lines 245-246 ) or “Under the irradiation of 620 nm wavelength light with 5 V bias voltage, the light amplitude llumination was 20 μW cm-2…” (in page 9, lines 265-266) or “…synthesized low trap density of states .9×1010cm-3…” (in page 16, lines 487-488), and some others sound strange to me. In addition to revising the wording of some sentences, the inclusion of comparative tables would result in greater clarity.
Response: Thank you very much for the reminding. As suggested, we have carefully checked and revised our manuscript language in the modified version. To describe the characteristics of the devices clearly, comparative Table 1 and Table 2 have been added in the revised paper.
4. Along with this, I have a number of comments and criticisms about the manuscript:
1) Beginning with some general issues, in my opinion there are formal aspects that must be corrected; specifically, there are sentences that are not clear, as well as other sentences that I consider unnecessary; in addition, several graphic schemes seem too simple for a scientific review. For example, Figures 6, 9 or 11 do not provide relevant information for the readers of a scientific article. The authors say that Figure 2 is the“…schematic diagram of the p-i-n perovskite”, but is the basic scheme of any p-i-n structure. On the other hand, the drawings of some devices (such us phototransistor in Figure 5) are assigned to "perovskite devices", although they are general schematics.
Response: Thank reviewer for the kind reminding. As suggested, we have checked and revised the sentences carefully to make the sentences clearer and more rigorous. Meanwhile, the mentioned diagrams also have been revised to be more informative for reading in the resubmitted version.
2) Some general paragraphs, such as those devoted to the fundamentals of some phenomena, should be revised; for example, in page 3, subsection 2.2 Perovskite photodiode, the description of a PIN junction is somewhat repetitive. I suggest a structure beginning with the description (brief) of general characteristics of photodiodes based on PIN junctions and then, in more detail, highlighting the advantages of using perovskite materials. On the other hand, there are some sentences that are unnecessary (in the context of a scientific article). For example, in page 8, lines 219 and 220: “… 2D perovskite refers to perovskite materials with 2D geometric structures. In other words, perovskite materials have large planes and thin thickness…” Similar comments would be applicable to other sections of the manuscript. In my opinion the structure of quite a few paragraphs needs to be revised to make the manuscript easier to read.
Response: Thanks for the constructive suggestion. As reminded, we have highlighted the advantages of using perovskite materials more detailed in the revised manuscript, and deleted the wordy and lengthy contents about the general characteristics of photodiodes based on PIN junctions. In addition, some unnecessary sentences and paragraph structure also have been carefully revised as suggested in the modified version to make the manuscript more clear and easier for reader.
3) The repetition of similar sentences or analogous explanations should be avoided. For example, in section 3.1, page 6, the description of 0D perovskites is repeated almost verbatim in:
-Lines 158-160. “0D perovskites are composed of isolated metal halide octahedral anions or metal halide clusters, which are surrounded by organic or inorganic cations with excitons strongly confined to each octahedron [6].”
-Lines 177-179. “…the 0D perovskite has a special structure in which the metal halide octahedra or metal halide clusters are isolated by the surrounding organic or inorganic cations and the excitons are strongly confined by the octahedra”
Response: Thanks for the constructive suggestion. We have revised the repetition of similar sentences and analogous explanations carefully as suggested in the revised manuscript.
4) In page 8, line 248 and following, the description of narrowband photodetectors is redundant: “…narrowband photodetectors only respond to a narrow spectrum, collecting photogenerated carriers of specific wavelengths and suppressing photogenerated carriers of other wavelengths […] The principle of its narrowband charge collection mechanism is to realize the narrowband spectral response by only collecting the photogenerated carriers at specific wavelengths and inhibiting the current generated by photogenerated carriers at other wavelengths. [48].
Response: Thank reviewer for the kind suggestion. We have deleted these sentences.
5) Page 12, lines 388 and 389, the definition of absorbance A is wrong (the right expression is A=log(Io/It). On the other hand, I suggest using the word “transmitted” instead of “exited” light intensity.
Response: Thanks for the kind reminding. We have modified the formula to A=logIo/logIt, and the word “exited” has been replaced to be “transmitted” as suggested.
6) The use of references in the final section (Conclusion and Outlook) is not appropriate.
Response: Thank reviewer for the kind suggestion. We have revised it in the modified version as suggested.
7) In addition to these aspects, there are a series sentences that need to be clarified/rewritten. Below, I list a few examples that can be read in the first pages, but the manuscript should be revised in depth; misprints must be corrected as well.
a) Page 1, in the Introduction, last sentence of the first paragraph. Is the word "As" missing at the beginning of the sentence? “As for quantum dot photodetectors that have developed rapidly in recent years, although they can be processed in solution…”. The first sentence of the second paragraph should be modified “With the development of perovskite materials, Due to its high carrier mobility, long carrier diffusion length, adjustable band gap [1-3], and strong light absorption ability [4-6], perovskite materials ...”. In the last sentences of the Introduction section, the verb tenses must be revised; I don't think I understand the meaning of the last sentence: “We believed that the future perovskite photodetectors would make a qualitative leap in stability, response speed, detectivity, and other aspects. These would be realized industrialization and improved people's life.”
Response: Thank you very much for the reminding. We have checked and revised the language, sentences, grammar in our manuscript carefully as suggested by reviewer.
b) Page 7, line 185 and following; It can be read that “Here are three potential research methods…”, but the authors give a list of four methods.
Response: Thanks for the kind reminding. We have revised it as suggested.
c) There have been quite a few typos in the use of uppercase and lowercase letters throughout the manuscript (I indicate below a few examples, but there are many)
Page 2, line 62: “…MSM absorb incident photons, When the photon energy…”
Page 8, line 244: Van der Waals
Page 12, lines 388 and 389: “it” and “t” should be upper case (“It” and “T” respectively) and “Is” after epsilon must be lower case. (is).
Page 16, line 483: Cs
Response: Thank you very much for the reminding. We have checked and revised the typos in the revised version.
d) The meaning of the acronyms should be entered the first time the terms are used. For example, Near Infrared is already used in the Introduction, but NIR is not introduced until page 8; PET is mentioned in page 7, "and polyethylene terephthalate (PET)" in page 8.
Response: Thank reviewer for the kind suggestion. As suggested, we have modified it in the revised manuscript.
5. Considering the aforementioned, my overall assessment is not positive, and I do not recommend the publication of the manuscript in its current state.
Response: Thank you very much for your constructive comments and kind work on our manuscript. As suggested, we have revised the existing problems carefully to improve the quality of our manuscript as you expected.

Reviewer 2 Report
In this review article, the authors have summarized the perovskite materials and device structures for photodetectors and their applications. The article will be a good reference for the scientific communities working on developing perovskite photodetectors. The Reviewer supports the publication with the following recommendations.
· Authors should highlight in their introduction how their review is different compared to other published work on transparent electrodes for organic optoelectronics devices; a couple of examples of many other reviews available online
o https://pubs.acs.org/doi/10.1021/acsaelm.1c01349
o https://onlinelibrary.wiley.com/doi/10.1002/adma.201605242
o https://iopscience.iop.org/article/10.1088/1674-1056/27/12/127806/meta?casa_token=1DqN8w4Z5UsAAAAA:pMWMiR4N9ITB-p1yZaUmjLaFegCxxn24D2R5yVb0f_PRSFvv0pG_pMc9qkM1d5xdwNyvMEkHSwA
· In the introduction and abstract, the authors mention the future outlook and perspective for perovskite photodetector research; they should add a section before the conclusion.
· The present title of the article appears to be saying about processing techniques. Unlike the actual article, the Reviewer suggests realigning the title with the actual review, e.g., Perovskite-based photodetectors and their applications.
· The practical application involving photodetector requires stable device performance; hence it is suggested to add a more details on the stability issue and expand the stability section.
A definitive purpose for using perovskites for photodetector can be added initially.
Author Response
Response to Reviewers’ Comments
In this review article, the authors have summarized the perovskite materials and device structures for photodetectors and their applications. The article will be a good reference for the scientific communities working on developing perovskite photodetectors. The Reviewer supports the publication with the following recommendations.
1. Authors should highlight in their introduction how their review is different compared to other published work on transparent electrodes for organic optoelectronics devices; a couple of examples of many other reviews available online.
https://pubs.acs.org/doi/10.1021/acsaelm.1c01349
https://onlinelibrary.wiley.com/doi/10.1002/adma.201605242
https://iopscience.iop.org/article/10.1088/16741056/27/12/127806/meta?casa_token=1DqN8w4Z5UsAAAAA:pMWMiR4N9ITBp1yZaUmjLaFegCxxn24D2R5yVb0f_PRSFvv0pG_pMc9qkM1d5xdwNyvMEkHSwA
Response: Thank you very much for the constructive suggestion. In our manuscript, the work mainly focuses on the perovskite photodetector types, material structure, practical applications (such as imaging, optical communication, biological detection fields etc.), and the avenues of device performance improvement (such as lead-free materials, stability), while that of reported studies (Li et al. Research progress of high-sensitivity perovskite photodetectors: A review of photodetectors: noise, structure, and materials. ACS Applied Electronic Materials, 2022, 4, 1485–1505; Ahmadi et al. A review on organic-inorganic halide perovskite photodetectors: Device engineering and fundamental physics. Advanced Materials, 2017, 29, 1605242; Zhao et al. Recent research process on perovskite photodetectors: A review for photodetector-materials, physics, and applications. Chinese Physics B, 2018, 27, 127806) mainly focuses on device structure design, interface engineering, fabrication approaches, and the physical mechanism of materials and devices. The above studies also have been cited in the revised manuscript as suggested.
2. In the introduction and abstract, the authors mention the future outlook and perspective for perovskite photodetector research; they should add a section before the conclusion.
Response: Thank reviewer for the kind reminding. We have added a special section about the outlook and perspective for perovskite photodetector research before the conclusion part as suggested.
3. The present title of the article appears to be saying about processing techniques. Unlike the actual article, the Reviewer suggests realigning the title with the actual review, e.g., Perovskite-based photodetectors and their applications.
Response: Thank reviewer for the constructive advice. To be clearer and more rigorous, we have revised the title to be “A review of perovskite-based photodetectors and their applications” as suggested.
4. The practical application involving photodetector requires stable device performance; hence it is suggested to add a more details on the stability issue and expand the stability section.
Response: Thank you very much for the constructive suggestion. We have added more details in the stability section in the revised manuscript. And some excellent studies have been cited in the revised version, such as “McMeekin et al. A mixed-cation lead mixed-halide perovskite absorber for tandem solar cells. Science, 2016, 3351, 151-155”; “Saliba et al. Incorporation of rubidium cations into perovskite solar cells improves photovoltaic performance. Science, 2016, 354, 206-209”; “Chen et al. Toward long-term stability: Single-crystal alloys of cesium-containing mixed cation and mixed halide perovskite. Journal of the American Chemical Society, 2019, 141, 1665-1671”; “Wu et al. Ultrasensitive, flexible perovskite nanowire photodetectors with long-term stability exceeding 5000 h. InfoMat 2022, 4, e12320”; “Zhang et al. Rubidium iodide-doped Spiro-OMeTAD as a hole-transporting material for efficient perovskite photodetectors. The Journal of Physical Chemistry C, 2022”.
5. A definitive purpose for using perovskites for photodetector can be added initially.
Response: Thank reviewer for the kind suggestion. It is well known that the complex and expensive manufacturing process is the tricky issue for the further development of existed Si-based, GaP, PbS photodetectors, while the difficult synthesis process, low light absorption capacity, and poor electric performance may be the fatal factors for polymer and colloidal quantum dot photodetectors. For perovskite material, the excellent electric characterizations (such as high carrier mobility, long carrier diffusion length), good optical performances (such as light absorption capacity, and tunable band gap), especially the simple and low-cost synthesis process make it become a promising candidate for the photodetector application. To be clearer, we have state above in the revised manuscript as suggested.

Reviewer 3 Report
The manuscript ID nanomaterials-2030991 is a review that mainly presents a study about particular perovskite structures with potential applications in photodetection. Photoconductors, photodiodes, and phototransistors are discussed. Please see below a list of comments to the authors:
1. A graphical abstract illustrating in the introduction section the main considerations analyzed in this work would be useful for readers.
2. A roadmap of the representative applications studies for the topic of this review is requested.
3. The presentation of the references in the panoramic description of the topic for the introduction could be improved in the collective form in the citations is split in individual form to better justify the selected publications to mention in this work.
4. Advantages and disadvantages of parameters associated with perovskite photoconductors, photodiodes, and phototransistors are suggested to be presented in a table for future discussions.
5. The authors are invited to discuss about potential perspective of perovkites sensitive to polarization and/or working with structured light. You can see for instance: https://doi.org/10.1016/j.optlastec.2021.107015
6. The abstract should clearly state what this works adds to literature in respect to other reviews or original papers published in the topic. You can see for instance: https://doi.org/10.1021/acsaelm.1c01349
7. Figure 10 should be improved in order to represent what is described in section 4.4.
8. The keywords should be improved in order to improve the expectation that this work can be found and cited for researchers reading the topic presented.
9. I noticed that only two 2022 references were included in the bibliography. It is suggested to update some citations to better support the cutting-edge information about the topic.
10. The data of some references should be verified and completed.
Author Response
Response to Reviewers’ Comments
1. A graphical abstract illustrating in the introduction section the main considerations analyzed in this work would be useful for readers.
Response: Thank you very much for the suggestion. As suggested, we have added a graphical abstract as shown below (as seen in attachment).
2. A roadmap of the representative applications studies for the topic of this review is requested.
Response: Thank you for the constructive advice. In the revised manuscript, we have added a roadmap as shown below (as seen in attachment).
3. The presentation of the references in the panoramic description of the topic for the introduction could be improved in the collective form in the citations is split in individual form to better justify the selected publications to mention in this work.
Response: Thank reviewer for the kind reminding. In the revised manuscript, we have sorted out the references and divided them according to the collective form as suggested.
4. Advantages and disadvantages of parameters associated with perovskite photoconductors, photodiodes, and phototransistors are suggested to be presented in a table for future discussions.
Response: Thank you very much for the kind reminding. As suggested, a table named Table 1 has been added in the revised manuscript to list the performance parameters of perovskite photoconductors, photodiodes, and phototransistors.
5. The authors are invited to discuss about potential perspective of perovskite sensitive to polarization and/or working with structured light. You can see for instance: https://doi.org/10.1016/j.optlastec.2021.107015
Response: Thank reviewer for the constructive suggestion. In the section 4.5, we introduced the application of some perovskite photodetectors in the field of polarized light. Moreover, some excellent studies have also been cited in the revised manuscript, such as (Hurtado-Aviles et al. Photo-induced structured waves by nanostructured topological insulator Bi2Te3. Optics & Laser Technology, 2021, 140, 107015; Song et al. Moiré Perovskite Photodetector toward High-Sensitive Digital Polarization Imaging. Advanced Energy Materials 2021, 11, 2100742; Zhang et al. Rational design of high-quality 2D/3D perovskite heterostructure crystals for record-performance polarization-sensitive photodetection. National Science Review, 2021, 8; Wang et al. A Chiral Reduced-Dimension Perovskite for an Efficient Flexible Circularly Polarized Light Photodetector. Angewandte Chemie International Edition 2020, 59, 6442-6450)
6. The abstract should clearly state what this works adds to literature in respect to other reviews or original papers published in the topic. You can see for instance: https://doi.org/10.1021/acsaelm.1c01349.
Response: Thank you very much for the kind reminding. In our manuscript, the work mainly focuses on the perovskite photodetector types, material structure, practical applications (such as imaging, optical communication, biological detection fields etc.), and the avenues of device performance improvement (such as lead-free materials, stability), while that of reported studies (Li et al. Research progress of high-sensitivity perovskite photodetectors: A review of photodetectors: noise, structure, and materials. ACS Applied Electronic Materials, 2022, 4, 1485-1505; Ahmadi et al. A review on organic-inorganic halide perovskite photodetectors: Device engineering and fundamental physics. Advanced Materials, 2017, 29, 1605242; Zhao et al. Recent research process on perovskite photodetectors: A review for photodetector-materials, physics, and applications. Chinese Physics B, 2018, 27, 127806) mainly focuses on device structure design, interface engineering, fabrication approaches, and the physical mechanism of materials and devices. To be more clearly, we have state above in the abstract and introduction as reviewer suggested. In addition, the above mentioned studies also have been cited in the revised manuscript as advice.
7. Figure 10 should be improved in order to represent what is described in section 4.4.
Response: Thank you very much for the reminding. We have modified Figure 10 (now Figure 11) in the revised manuscript as suggested.
8. The keywords should be improved in order to improve the expectation that this work can be found and cited for researchers reading the topic presented.
Response: Thank reviewer for the kind reminding. We have changed the keywords to be: perovskite; photodetector; optical detection; lead free.
9. I noticed that only two 2022 references were included in the bibliography. It is suggested to update some citations to better support the cutting-edge information about the topic.
Response: Thanks for the constructive suggestion. We have added some 2022 references in the revised version as reviewer suggested.
10. The data of some references should be verified and completed.
Response: Thank reviewer for the constructive suggestion. We have checked and verified the data form the cited references carefully as suggested.

Round 2
Reviewer 1 Report
In my opinion, the revised manuscript has been significantly improved compared to the original version. However, there are still a number of issues that the authors need to correct. Specifically, there are some sentences that need to be rewritten, and there are still a number of grammatical errors and typos. I list a few below (with more details for the first few pages) but authors should review the entire manuscript carefully.
1. Page 2, line 62: In the title of the section “…Perovskite photodetector Type…” the word “type” should be lowercase.
2. Page 2, line 67: The sentence “Perovskite photoconductive detectors deposit metal electrodes at both ends …” must be rewritten.
3. Page 2, line 69: In the sentence “…the electrons of MSM absorbs enough…” the verb must be “absorb” (without the “s”).
4. Page 3, line 90: It can be read that “The structural schematic diagram of the p-i-n type photodetector is shown in Figure 3”, but (according to the caption for figure 3), the figure represents a p-i-n photodiode.
5. Page 3, line 93: “…machanical” must be changed by “mechanical”.
6. Some inconsistencies in size and font in some added paragraphs need to be corrected, such as in the new caption for figure 4 (page 4). On the other hand, the resolution of the labels in that figure is not high enough.
7. Page 5, line 150: the title of section 3 (Material-dimensional perovskite photodetectors) sounds strange to me.
8. Pages 6, (line 187) and page 13 (line 419), Stokes must be uppercase.
9. Pages 6 and 7, line 189 and following. I have some comments about the list of strategies to adjust the perovskite band gap:
a) In my opinion, the quite general sentence “Perovskites are ABX3 structure, where A is monovalent …. halide anions (I-, Br-, Cl-)” is not in the right place.
b) Upper case must be used after the point.
c) A common writing style should be used in all four strategies; if the gerund verb form (“Replacing…”, “Trying…”) is used to introduce each strategy, it should also be used in the first one.
10. Page 8, line 268. “Low” should be lowercase, low.
11. Page 12, lines 387 and 388, the definition of absorbance A is still wrong. The right expression for absorbance is A=log(Io/It), not log(Io)/log(It). In addition, “…T is transmittance.,” instead of “…t is transmittance...,” must be written.
12. Page 14, I would suggest the title "Polarized light imaging" for the section 4.5.
13. Page 17, line 543: The sentence “For the problem of improving the stability of perovskite, an effective method is to improve the stability by controlling the composition and…” can be simplified: “In order to improve the stability of perovskite, an effective method is controlling the composition and…”
As I said at the beginning of my comments, in addition to correcting the aforementioned errors, I highly recommend that authors review all text for grammatical errors and typos.
Author Response
Response to Reviewers’ Comments
In my opinion, the revised manuscript has been significantly improved compared to the original version. However, there are still a number of issues that the authors need to correct. Specifically, there are some sentences that need to be rewritten, and there are still a number of grammatical errors and typos. I list a few below (with more details for the first few pages) but authors should review the entire manuscript carefully.
1. Page 2, line 62: In the title of the section “…Perovskite photodetector Type…” the word “type”should be lowercase.
Response: Thank you very much for the reminding. We have revised it in the revised version.
2. Page 2, line 67: The sentence “Perovskite photoconductive detectors deposit metal electrodes at both ends …” must be rewritten.
Response: Thank reviewer for the kind reminding. To be clear and rigorous, we have revised this sentence to be “Metal electrodes were deposited on the perovskite surface to form a metal-semiconductor-metal (MSM) coplanar structure. The material for mental electrodes includes Au, Ag, Al, Cu, and other metals.” in the revised manuscript.
3. Page 2, line 69: In the sentence “…the electrons of MSM absorbs enough…” the verb must be “absorb” (without the “s”).
Response: Thank reviewer for the kind reminding. We have modified it in the revised manuscript.
4. Page 3, line 90: It can be read that “The structural schematic diagram of the p-i-n type photodetector is shown in Figure 3”, but (according to the caption for figure 3), the figure represents a p-i-n photodiode.
Response: Thanks for the kind reminding. We have checked and revised in the resubmitted version.
5. Page 3, line 93: “…machanical” must be changed by “mechanical”.
Response: Thank you very much for the reminding. We have checked and revised this in the revised version.
6. Some inconsistencies in size and font in some added paragraphs need to be corrected, such as in the new caption for figure 4 (page 4). On the other hand, the resolution of the labels in that figure is not high enough.
Response: Thank reviewer for the constructive suggestion. We have checked the whole manuscript and revised the size and font. Moreover, the figure has been replaced by a higher resolution one.
7. Page 5, line 150: the title of section 3 (Material-dimensional perovskite photodetectors) sounds strange to me.
Response: Thanks for the constructive suggestion. We have revised the title of section 3 to “Perovskite photodetectors with low-dimensional materials” to make the manuscript clear and more rigorous.
8. Pages 6, (line 187) and page 13 (line 419), Stokes must be uppercase.
Response: Thank you very much for the reminding. We have revised it in the revised version.
9. Pages 6 and 7, line 189 and following. I have some comments about the list of strategies to adjust the perovskite band gap:
a) In my opinion, the quite general sentence “Perovskites are ABX3 structure, where A is monovalent …. halide anions (I-, Br-, Cl-)” is not in the right place.
Response: Thank reviewer for the constructive advice. To be clearer and more rigorous, we moved this sentence to the Introduction part in the modified version.
b) Upper case must be used after the point.
Response: Thanks for the kind reminding. We have revised it as suggested.
c) A common writing style should be used in all four strategies; if the gerund verb form (“Replacing…”, “Trying…”) is used to introduce each strategy, it should also be used in the first one.
Response: Thank you very much for the constructive suggestion. We have revised it as suggested.
10. Page 8, line 268. “Low” should be lowercase,low.
Response: Thanks for the kind reminding. We have revised it as suggested.
11. Page 12, lines 387 and 388, the definition of absorbance A is still wrong. The right expression for absorbance is A=log(Io/It), not log(Io)/log(It). In addition, “…Tis transmittance.,” instead of “…t is transmittance...,” must be written.
Response: Thank reviewer for the kind suggestion. We have corrected the expression of absorbance A. Moreover, the sentence “…t is transmittance” has been replaced by “…T is transmittance” as suggested.
12. Page 14, I would suggest the title "Polarized light imaging" for the section 4.5.
Response: Thank reviewer for the constructive advice. We have modified the title of section 4.5 to be “Polarized light imaging” in the revised version as suggested.
13. Page 17, line 543: The sentence “For the problem of improving the stability of perovskite, an effective method is to improve the stability by controlling the composition and…” can be simplified: “In order to improve the stability of perovskite, an effective method is controlling the composition and…”
Response: Thank you for the constructive advice. We have simplified this sentence in the revised manuscript as suggested.
14. As I said at the beginning of my comments, in addition to correcting the aforementioned errors, I highly recommend that authors review all text for grammatical errors and typos.
Response: Thank you very much for your constructive comments. As suggested, we have checked and revised the manuscript carefully to improve the grammatical errors and typos.

Reviewer 3 Report
The manuscript presents a very interesting study with fundamental informaction that could be a base for future research. The reviewed version of the manuscript fulfills the standards of this prestigious journal, and a good impact of this manuscript can be expected in the field of perovskite structures with potential applications in photodetection. I can recommend this work for publication in present form.
Author Response
Response to Reviewers’ Comments
The manuscript presents a very interesting study with fundamental informaction that could be a base for future research. The reviewed version of the manuscript fulfills the standards of this prestigious journal, and a good impact of this manuscript can be expected in the field of perovskite structures with potential applications in photodetection. I can recommend this work for publication in present form.
Response: Thank reviewer for the kind comments. We sincerely thank the reviewer for all the help on our manuscript.
